# Supply Chain Coordination under Carbon Emission Tax Regulation Considering Greening Technology Investment

**DOI:** 10.3390/ijerph19159232

**Published:** 2022-07-28

**Authors:** Zhimin Wei, Yun Huang

**Affiliations:** 1School of Business, Macau University of Science and Technology, Macao, China; zhiminweigxzysfxy@163.com; 2School of Economics and Trade, Guangxi Vocational Normal University, Nanning 530007, China

**Keywords:** supply chain coordination, greening technology investment, advance purchase discount contract, option contract, carbon emission tax regulation

## Abstract

In this paper, we firstly derive the optimal strategies, including greening technology investment, production volume and order quantity decisions with stochastic demand, for the emissions-dependent supply chain composed of one manufacturer and one retailer. Then, an advance purchase discount (APD) contract and an option contract are applied to coordinate the supply chain. Moreover, an innovative prepayment-based option (PBO) contract is designed based on an APD contract and an option contract. We discuss the cash flow, the inventory risk allocation and the impacts of carbon emission tax under each contract. It is found that considering improving cash flow, preselling (or option selling) as a means of supporting the manufacturer with sufficient cash flow will help expand production and invest in greening technology. From the perspective of avoiding inventory risk, the APD contract benefits the manufacturer while the option contract benefits the retailer. However, the PBO contract generates intermediate allocations of inventory risk between manufacturer and retailer.

## 1. Introduction

Many countries have developed national carbon policies since the proposal of the “low-carbon economy” in the late 20th century. The carbon emission tax is one of the critical carbon policies [1] and an effective means adopted by many countries for economic development and environmental improvement [2,3]. However, carbon emission tax results in higher companies’ operations cost, especially when output increases [4]. Therefore, considering carbon emission tax regulation, it is a new challenge to manage a supply chain in a cost-effective way [5].

Many scholars focused on optimizing supply chain decisions under carbon emission tax regulation with deterministic demand [6,7,8,9,10,11]. In practice, supply chain partners often face the risk of overstocking or understocking caused by the uncertainty of demand [12]. However, to reduce tax burdens, the manufacturer has an incentive to invest in greening technology [13]. Specifically, there may be significant financial pressure if the upstream manufacturer invests in production and greening technology when demand is still uncertain [14]. It might also cause insufficient or even interrupted supply to a downstream retailer with a capital-constrained manufacturer [15]. Few researchers have explored operational management with stochastic demand in the supply chain [9,16,17], let alone considered reducing carbon emission by investing in greening technology simultaneously. Therefore, we focus on how to reduce the risks caused by uncertain demand and alleviate the financial pressure caused by greening technology investment and carbon emission tax in supply chain management.

Based on supply chain finance theory, external financing models and internal financing models can be used to alleviate financial pressure. The external financing model means that enterprises obtain credit financing from banks and other financial institutions outside the supply chain. The internal financing model states that enterprises obtain trade financing from supply chain partners, mainly in two ways, i.e., deferred payment and prepayment. Ref. [18] have proved that prepayment could ease the financial pressure of the supply chain. A contract mechanism is an effective way to achieve supply chain coordination, which can solve the problems of risk-sharing, information asymmetry and profit distribution. An advance purchase discount (APD) contract could better realize risk-sharing of partners within a supply chain [19]. The retailer can obtain a discount by prepayment, but they have to bear inventory risk when actual demand is less than the preorder quantity. Ref. [20] studied the impact of an APD contract on total cost and carbon emissions under carbon emission tax regulation. Meanwhile, an option can be used to hedge risks of demand uncertainty in supply chain management [21]. The retailer has the right to determine the number of options exercised based on realized demand. The manufacturer has to take inventory risk into full consideration. Ref. [22] proved that an option contract could decrease demand risk and inventory costs. Ref. [23] studied the optimal decisions of a carbon-constrained manufacturer with option contract under stochastic demand. These researchers only study the application of the APD contract or the option contract alone. However, a single contract has difficulty in optimizing the supply chain, when simultaneously considering releasing financial pressure and hedging the risks of demand uncertainty. Thus, we innovatively designed a prepayment-based option (PBO) based on an APD contract and an option contract to coordinate a manufacturer–retailer supply chain under carbon emission tax regulation considering greening technology investment and stochastic demand.

In practice, many emission-dependent industries, such as energy, manufacturing and electronics, alleviate the financial pressure with the APD contract and hedge market risk with the option contract [24]. For example, the option contract is employed in Hewlett-Packard’s purchasing of memory chips [25], and the “unearned revenue” (i.e., advance payment) of Zhuhai Gree Corporation had reached RMB 10 billion in 2016 [5]. Moreover, IBM’s printer business trades on an option contract and Midea’s manufacturers also ask retailers to pay in advance [25]. Under the highlights of carbon emission and carbon neutralization of different countries, it is important to study how to apply an APD contract, option contract or other contract for emission-dependent supply chain coordination, with a consideration of greening technology investment, carbon emission tax regulation and stochastic demand. Therefore, the following research questions are proposed: (1)With stochastic demand, how does the governmental carbon emission tax policy affect the decision making on the greening technology investment and other operational decision variables under decentralized and centralized supply chains?(2)When the manufacturer invests in greening technology, how do the APD contract, the option contract and the PBO contract improve cash flow and risk allocation of partners within the supply chain, under carbon emission tax regulation?(3)Under carbon emission tax regulation, what are different impacts of the APD contract, option contract and PBO contract on cash flow and risk allocation of supply chain partners?


To answer these questions, we firstly derive the optimal operational decisions, such as the production volume, the carbon emission reduction rate and the order quantity, considering greening technology investment and stochastic demand for decentralized and centralized supply chains under carbon emission tax regulation. Then, we discuss coordination mechanisms (i.e., the APD contract, the option contract and the PBO contract) for the emission-dependent supply chain. We also demonstrate that the supply chain can be coordinated with these contracts. Meanwhile, we study the cash flow, the inventory risk allocation and the influences of carbon emission tax under each contract.

On a broader level, this paper makes the following contributions. Firstly, we derive the optimal strategies, including greening technology investment, production volume and order quantity decisions with stochastic demand for decentralized and centralized emission-dependent supply chains composed of one manufacturer and one retailer. Secondly, we apply an advance purchase discount (APD) contract and an option contract to coordinate the emission-dependent supply chain. Lastly, a novel prepayment-based option (PBO) contract is designed, which is based on an APD contract and an option contract. It is found that for improving cash flow, preselling (or option selling), as a means of raising funds, provides the manufacturer with sufficient cash flow. Then, it is able to expand production and invest in greening technology. Meanwhile, both capacity and greening technology investment can be increased when the discount factor or the option price reaches a certain threshold. From an avoiding inventory risk perspective, the APD contract benefits the manufacturer while the option contract benefits the retailer. However, the PBO contract generates intermediate allocations of inventory risk between manufacturer and retailer. For the impacts of governmental carbon emission tax policy, it does not influence the optimal order decision of the retailer. However, the optimal carbon emission reduction rate of the manufacturer and the carbon emission tax are positively related.

The remainder of this paper is organized as follows. In Section 2, we review the relevant literature. The problem set is described in Section 3. In Section 4, we analyze the decentralized supply chain and the centralized supply chain. Section 5 studies supply chain coordination. In Section 6, we discuss the results. Finally, we present the conclusion in Section 7.

## 2. Literature Review

Research related to this paper includes: (i) the optimal decision making and coordination mechanism of the supply chain under carbon emission tax regulation; (ii) the supply chain coordination with the APD contract and the option contract.

On the impact of carbon tax policy on an enterprise’s optimal decision making, Ref. [6] found that both retail price and wholesale price rise as carbon emission tax rises. Ref. [26] proved that a manufacturer could reduce production cost and carbon emission tax by recycling waste products from consumers. However, they did not discuss reducing carbon emissions by investing in greening technology. Ref. [27] studied the impact of carbon emission tax regulation on the manufacturer’s optimal decision making under two scenarios, i.e., no greening technology investment and with investment. They showed that manufacturers had an incentive to invest in greening technology under carbon emission tax regulation. Ref. [8] found that greening technology investment can improve environmental performance and supply chain coordination could be achieved by a revenue-sharing contract. Refs. [10,28] also proved that collaborative emission reduction could be achieved with a cost-sharing contract. Ref. [11] found that a modified cost-sharing contract could reach a win–win situation for retailer and supplier under carbon emission tax regulation. Ref. [7] indicated that there is a high carbon emission reduction rate with vertical cooperation while there is low consumers’ welfare with horizontal cooperation. Ref. [29] studied the manufacturer’s optimal carbon emission reduction rate under carbon emission policy considering cournot competition and collusion. Most studies focus on green supply chain coordination based on deterministic market demand. Ref. [9] studied the demand for uncertainty, and they found that centralized decision making is conducive to maximizing the total profit of a supply chain and achieve the government’s goals of carbon emission reduction. Ref. [30] found that the manufacturer’s optimal decision making was influenced by consumer product acceptance and carbon emission tax regulation. However, the studies of [9,30] are limited to a uniform distribution demand. Different from the above literature, both greening technology investment and stochastic demand are studied in our supply chain coordination problem. Meanwhile, relieving the financial pressures of investing in greening technology is another consideration of our study.

Many researchers have studied how an APD contract and an option contract coordinate the supply chain. Ref. [31] found that the APD contract benefited a capital-constrained supply chain. Ref. [32] concluded that an APD contract could provide significant Pareto improvement for the supply chain. Moreover, ref. [33] explored how to make the inventory decision under the APD discount contract. Ref. [34] studied how a newsvendor retailer made a preselling strategy considering the loss-averse consumer. They found that retailers would use a deep preselling strategy with goods with a greater profit margin. Ref. [35] indicated that the retailer would adopt a preselling strategy when the consumers’ expected surplus value was higher. Moreover, the preselling strategy was influenced by market parameters and consumers. Ref. [15] proved that the prepaying mode could help supply chain parties achieve a win–win situation, especially for the capital-constrained supplier. Compared with a bank loan, the supplier was willing to obtain advance payment from the retailer [36,37]. Ref. [38] studied the optimal green remanufacturing production decisions under the prepayment model. Subsequently, scholars focused on how to utilize prepayment to solve the problem of insufficient environmental protection funds in a supply chain. Ref. [39] proved that the prepayment could help small and medium-sized enterprises of the supply chain to release the financial pressure of investing in greening technology. Ref. [38] worked out the optimal prepaid ratio required in the green supply chain. However, the impact of carbon policy was not discussed in their model.

On the other hand, ref. [21] showed that the supply chain coordination could be reached with an option contract when exercise price and the cost coefficient were linearly related. In addition to achieving supply chain coordination, ref. [40] proved that a put option could decrease the uncertainty of the retailer’s profit margins under stochastic demand. Ref. [41] indicated that the total profit of the supply chain with an option contract was more than that without a contract. Ref. [42] considered the loss-averse supply chain coordination with an option contract. However, their studies are limited to a price-dependent demand. Ref. [43] proved that the option contract could coordinate both the supplier-led and the retailer-led supply chains under stochastic market demand. Ref. [44] found that the manufacturer’s optimal decision was influenced by their level of overconfidence, under the option contract. An option contract has been demonstrated to coordinate the relief supply chain [45], the fresh food supply chain [46], etc. Although APD contracts and option contracts have been widely used in supply chain management, they were rarely used in green supply chain setting. We introduced an APD contract and an option contract into the green supply chain, under carbon emission tax regulation and considering greening technology investment and financial pressure. Moreover, in order to optimize the supply chain in complex situations considering both relieving financial pressure and hedging the risks from demand uncertainty, we coordinate the green supply chain with a newly proposed PBO contract, which combines an APD contract with an option contract. Currently, few studies focus on the combination of APD contracts and option contracts to study supply chain coordination in such complex situations. In this paper, we discuss how to achieve better risk-sharing and improve cash flow from the perspective of an overall emission-dependent supply chain under different contracts.

## 3. Model Description and Hypotheses

We study the green supply chain coordination with contracts in a two-echelon supply chain, consisting of one retailer (he) and one manufacturer (she). The notations used are listed in Table 1. The following hypotheses are proposed:(a)The core of the carbon emission tax regulation is tax or price-based regimes. Tax rate is set by the government. The carbon emission tax function is an increasing function of the total amount of emission [14]. Enterprise pays the cost of tax for carbon emission.(b)Carbon emission is considered in the manufacturer’s production stage [10]. To reduce tax burdens, the manufacturer can reduce carbon emission by investing in greening technology and determine the carbon emission reduction rate. It is assumed that the greening technology investment cost is a quadratic function of carbon emission reduction rate, i.e., $\frac{1}{2}\xi \Delta e^{2}$ [8,14].(c)The stochastic demand x has a risk-neutral equivalent cumulative distribution function F(x), a probability density function f(x), with mean value of *µ* and variance of $\sigma^{2}$ [47].(d)In equilibrium, supply chain partners have positive demands and profits [41].


**Table 1 ijerph-19-09232-t001:** Parameters and decision variables.

**(a) Notations for the manufacturer**	
Q	The manufacturer’s production volume under the decentralized system (decision variable)
$Q_{A}$	The manufacturer’s production volume with the APD contract (decision variable)
$Q_{O}$	The manufacturer’s production volume with the option contract (decision variable)
$Q_{M}$	The manufacturer’s production volume with the PBO contract (decision variable)
Δe	The carbon emission reduction rate under the decentralized system (decision variable)
$\Delta e_{A}$	The carbon emission reduction rate with the APD contract (decision variable)
$\Delta e_{O}$	The carbon emission reduction rate with the option contract (decision variable)
$\Delta e_{M}$	The carbon emission reduction rate with the PBO contract (decision variable)
e	The initial carbon emission per unit output
ξ	The coefficient of greening technology investment cost
t	The tax rate per unit carbon emission
r	The wholesale price discount factor, 0 < *r* < 1
w	The wholesale price per unit, $c_{m}+te\left(1-\Delta e\right)<\frac{w}{\left(1\;+\;r\right)}<w$
$c_{0}$	The option price per unit
$c_{e}$	The strike price per unit, $c_{m}+te\left(1-\Delta e\right)<c_{0}+c_{e}<w$
$c_{m}$	The production cost per unit
e	The initial carbon emission per unit product
$\Pi_{m}$	The total profit of manufacturer
**(b) Notations for the retailer**	
$q_{A}$	The retailer’s preorder quantity with the APD contract (decision variable)
$D_{A}$	The prepayment with the APD contract, $D_{A}=\frac{q_{A}w}{\left(1\;+\;r\right)}$
$q_{O}$	The retailer’s option order quantity with the option contract (decision variable)
$y_{A}$	The retailer’s preorder quantity with the PBO contract (decision variable)
$y_{O}$	The retailer’s option order quantity with the PBO contract (decision variable)
p	The retail price per unit
v	The salvage value per unit, *p* > *w* > *c_m_* + *te*(1 − Δ*e*) > *v*
$\Pi_{r}$	Total profit of the retailer
**(c) Notations for the supply chain**	
$Q_{c}$	The manufacturer’s production volume in the centralized system (decision variable)
$\Delta e_{c}$	The carbon emission reduction rate in the centralized system (decision variable)
$\Pi_{c}$	The total profit of supply chain in the centralized system
**(d) Notations** **for the timeline**	
$T_{0}$	Before the selling season
$T_{1}$	During the selling season
$T_{2}$	At the end of selling season

The game timeline of manufacturer and retailer is as follows (see Figure 1):

Stage 1: Manufacturer announces the wholesale price w, the discount factor r, the option price $c_{0}$ and the strike price $c_{e}$.

Stage 2: Retailer makes an order. Retailer determines the prepayment $D_{A}$ (i.e., the preorder quantity $q_{A}=\frac{D_{A}\left(1\;+\;r\right)}{w}$) of APD contract, the option order quantity $q_{O}$ of option contract, the preorder quantity $y_{A}$ and the option order quantity $\;y_{O}$ of PBO contract.

Stage 3: Manufacturer determines the production volume $\left(Q_{A},Q_{O},Q_{M}\right)$ and the carbon emission reduction rate $\left(\Delta e_{A},\Delta e_{O},\Delta e_{M}\right)$. She is required to guarantee the preorder quantity and the option order quantity. Therefore, her production volume satisfies the constraints $Q_{A}\geq q_{A},\;\;Q_{O}\geq q_{O},\;\;Q_{M}\geq y_{A}+y_{O}$ in the above cases.

Stage 4: Retailer could place a second order at the unit price w until he has observed actual demand  x [48]. That is, he buys products once more if his order quantity could not satisfy market demand (i.e., $\;q_{A}<x$ under the APD contract, $q_{O}<x$ under the option contract, $y_{A}+y_{O}<x$ under the PBO contract). In addition, the retailer firstly determines how to exercise options based on realized demand under the option contract. Moreover, there are three scenarios with the PBO contract. If actual demand is lower than the preorder quantity of the retailer $\left(x<y_{A}\right)$, he does not choose to exercise options. If actual demand is higher than his preorder quantity and lower than his total order quantity $\left(y_{A}<x<y_{A}+y_{O}\right)$, the retailer will choose to exercise some of the options. If the total order quantity is lower than actual demand $\;(y_{A}+y_{O}<x)$, the retailer not only exercises all options but places a second order.

## 4. Model Formulation

### 4.1. The Decentralized Supply Chain

Firstly, the manufacturer determines the production volume Q and the carbon emission reduction rate  Δe at ${\;T}_{0}$ because she has to organize production before demand is realized. Then, the retailer places an order until he has observed actual demand  x at $T_{1}$. If x<Q, the order quantity of the retailer is x and his transfer payment is wx. Meanwhile, inventory risk is borne by the manufacturer. If x>Q, the order quantity of the retailer is Q and his transfer payment is wQ. Meanwhile, the stock shortage risk is borne by the retailer. That means whether the retailer can order enough products to meet market demand or not depends on the production volume of the manufacturer. Therefore, total profit of manufacturer will be
(1)$$\Pi_{m}\left(Q,\Delta e\right)=wS\left(Q\right)+v\left[Q-S\left(Q\right)\right]-c_{m}Q-te\left(1-\Delta e\right)Q-\frac{1}{2}\xi {\left(\Delta e\right)}^{2}.$$

In the above profit function, $\;S\left(Q\right)=Q-\int_{0}^{Q}F\left(x\right)dx$ represents the expected sales and  Q−S(Q) represents the expected inventory. The first term is total sales revenue, the second term is total salvage value, the third term is total production cost, the fourth term is cost of carbon emission tax and the last term is cost of investing in greening technology.

**Proposition** **1.***In the decentralized system, the expected profit of the manufacturer $\Pi_{m}\left(Q,\Delta e\right)$ is concave in Q and Δe. There exists an optimal production volume $Q^{*}$ and an optimal carbon emission reduction rate $\;\Delta e^{*}$ under the decentralized system. Furthermore, it satisfies the following conditions: $\left[w-c_{m}-te\left(1-\Delta e^{*}\right)\right]-\left(w-v\right)F\left(Q^{*}\right)=0$, $\Delta e^{*}=\frac{teQ^{*}}{\xi }$ and $f\left(Q^{*}\right)>\frac{{\left(te\right)}^{2}}{\xi \left(w\;-\;v\right)}$*.

Proof. See Appendix A. 

From Proposition 1, we know that the optimal production volume $Q^{*}$ increases with wholesale price w while it decreases with total manufacturing cost $\;\left[c_{m}+te\left(1-\Delta e^{*}\right)\right]$, i.e., $\;\frac{\partial Q^{*}}{\partial w}>0$, $\frac{\partial Q^{*}}{\partial \left[c_{m}\;+\;te\left(1\;-\;\Delta e^{*}\right)\right]}<0$. That means raising the wholesale price or lowering the total manufacturing cost can increase the optimal production volume of the manufacturer.

Then, total expected profit of the retailer will be $\Pi_{r}=\left(p-w\right)\left[Q^{*}-\int_{0}^{Q^{*}}F\left(x\right)dx\right]=\left(p-w\right)S\left(Q^{*}\right)$. It is not difficult to find that the carbon emission tax will not influence the total profit of the retailer. However, the optimal carbon emission reduction rate of the manufacturer and the carbon emission tax are positively related, i.e., $\frac{\partial \Delta e^{*}}{\partial t}>0$.

### 4.2. The Centralized Supply Chain

The manufacturer and retailer jointly decide the production volume and the carbon emission reduction rate because they work jointly as a unified enterprise in the centralized system. Then, total expected profit of overall supply chain will be
(2)$$\Pi_{c}\left(Q_{c},\Delta e_{c}\right)=pS\left(Q_{c}\right)+vI\left(Q_{c}\right)-c_{m}Q_{c}-tQ_{c}e\left(1-\Delta e_{c}\right)-\frac{1}{2}\xi \Delta e_{c}{}^{2}.$$

In the above profit function, $\;S\left(Q_{c}\right)=Q_{c}-\int_{0}^{Q_{c}}F\left(x\right)dx$ represents the expected sales and $I\left(Q_{c}\right)=Q_{c}-S\left(Q_{c}\right)$ represents the expected inventory. The first term is total sales revenue, the second term is total salvage value, the third term is total production cost, the fourth term is cost of carbon emission tax, the last term is cost of investing in greening technology. Note that the cost of ordering products at wholesale price is not included in the above representation, because the ordering cost is merely the income transferred between retailer and manufacturer.

**Proposition** **2.**
*In the centralized system, the expected profit of the overall supply chain $\Pi_{c}\left(Q_{c},\Delta e_{c}\right)$ is concave in $Q_{c}$ and $\Delta e_{c}$. There exists an optimal production volume $Q_{c}^{*}$ and an optimal carbon emission reduction rate $\;\Delta e_{c}^{*}$, under the centralized system. Furthermore, it satisfies the following conditions: $\left[p-c_{m}-te\left(1-\Delta e_{c}^{*}\right)\right]-\left(p-v\right)F\left(Q_{c}^{*}\right)=0$, $\Delta e_{c}^{*}=\frac{teQ_{c}^{*}}{\xi }$ and $f\left(Q_{c}^{*}\right)>\frac{{\left(te\right)}^{2}}{\xi \left(p\;-\;v\right)}$.*


Proof. See Appendix A.

From Proposition 2, we find that the optimal production volume and the optimal carbon emission reduction rate under the centralized system are greater than those under the decentralized system, i.e., $Q_{c}^{*}>Q^{*}$ and $\Delta e_{c}^{*}>\Delta e^{*}$. That means the performance of the centralized system is better than that of the decentralized supply chain. Therefore, the performance of the decentralized system should be improved with a more attractive contract mechanism.

## 5. Supply Chain Coordination with Contracts

### 5.1. The Advance Purchase Discount (APD) Contract

Under the APD contract, firstly, the retailer optimizes preorder quantity $q_{A}^{*}\;$(i.e., prepayment $D_{A}^{*}=\frac{q_{A}^{*}w}{1\;+\;r}$). Then, the manufacturer optimizes production volume $Q_{A}^{*}$ and carbon emission reduction rate $\Delta e_{A}^{*}$. The cash flow of the supply chain with the APD contract is as follows (see Figure 2):

$T_{0}$: Retailer pays $D_{A}$ in advance.

$T_{1}$: $Q_{A}-\int_{0}^{Q_{A}}F\left(x\right)dx$ represents the retailer’s expected sales and his total sales revenue would be $p\left[Q_{A}-\int_{0}^{Q_{A}}F\left(x\right)dx\right]$. The retailer buys products once more if his preorder could not satisfy market demand. Meanwhile, the retailer’s expected transfer payment is $w\left[Q_{A}-\int_{q_{A}}^{Q_{A}}F\left(x\right)dx-q_{A}\right]$.

$T_{2}$: The retailer’s expected inventory is $q_{A}-\left(q_{A}-\int_{0}^{q_{A}}F\left(x\right)dx\right)=\int_{0}^{q_{A}}F\left(x\right)dx$ while the manufacturer’s expected inventory is $Q_{A}-\left(Q_{A}-\int_{q_{A}}^{Q_{A}}F\left(x\right)dx\right)=\int_{q_{A}}^{Q_{A}}F\left(x\right)dx$. The per-unit salvage value is v. Then, the manufacturer pays carbon emission tax $te\left(1-\Delta e_{A}\right)Q_{A}$.

Therefore, total profit of retailer will be
(3)$$\Pi_{r}\left(q_{A}\right)=p\left[Q_{A}-\int_{0}^{Q_{A}}F\left(x\right)dx\right]\;+\;v\int_{0}^{q_{A}}F\left(x\right)dx-w\left[Q_{A}-\int_{q_{A}}^{Q_{A}}F\left(x\right)dx-q_{A}\right]-D_{A}\;.$$

In Equation (3), the first term is total expected sales revenue, the second term is total salvage value, the third term is cost of a second order and the last term is a prepayment.

Total profit of manufacturer will be
(4)$$\begin{array}{c} \Pi_{m}\left(Q_{A},\Delta e_{A}\right)=D_{A}\;+\;w\left[Q_{A}-\int_{q_{A}}^{Q_{A}}F\left(x\right)dx-q_{A}\right]\;+\;v\int_{q_{A}}^{Q_{A}}F\left(x\right)dx\\ -\left(c_{m}\;+\;te\left(1-\Delta e_{A}\right)\right)Q_{A}-\frac{1}{2}\xi {\left(\Delta e_{A}\right)}^{2}, \end{array}$$
(5)$$S.t.q_{A}\leq Q_{A}.$$

Equation (4) includes the sales revenue from preselling the sales revenue from second-selling, the total salvage value, the total cost of production and carbon emission tax, as well as the cost of greening technology investment. The manufacturer is required to ensure the preorder quantity. Therefore, her production volume satisfies the constraint $Q_{A}\geq q_{A}$.

The advance purchase discount model is a financing method within the supply chain. The manufacturer obtains a short-term loan $D_{A}$ with an interest rate r from the retailer, and she is required to commit a minimum production volume $q_{A}$. When the retailer has observed the actual demand x, he purchases products with a quantity of $min\left(x,\;Q_{A}\right)$ at wholesale price w and pays the full amount. The manufacturer repays the retailer’s principal and interest $D_{A}\;\left(1\;+\;r\right)$. If the manufacturer is unable to repay it, the retailer receives the full revenue wx of the manufacturer.

According to the dynamic game theory, firstly, the manufacturer’s optimal decisions are solved considering the retailer’s strategy set, and then the retailer’s optimal decisions are solved based on the optimal strategy of the manufacturer [49]. Therefore, two cases are considered for the retailer’s strategy: (i) the preorder quantity of the retailer is lower than the production volume of the manufacturer, (ii) vice versa. Using the Lagrangian relaxation approach, the manufacturer’s optimal production volume is $Q_{A1}^{*}=Q_{A}^{*}=F^{-1}\left(\frac{w\;-\;c_{m}\;-\;te\left(1\;-\;\Delta e_{A}^{*}\right)}{w\;-\;v}\right)$ and her optimal carbon emission reduction rate is $\Delta e_{A1}^{*}=\Delta e_{A}^{*}=\frac{teQ_{A}^{*}}{\xi }\;$ when $\;q_{A}<Q_{A}^{*}$. However, the manufacturer’s optimal production volume is $\;Q_{A2}^{*}=q_{A}$ and her optimal carbon emission reduction rate is $\Delta e_{A2}^{*}=\frac{teq_{A}}{\xi }$ when $q_{A}\geq Q_{A}^{*}$. That means the manufacturer’s optimal production volume with the APD contract is $Q_{A}^{*}=max\left(q_{A},Q_{A}^{*}\right)$. This result is similar to that of [19], although his model did not consider carbon emission tax and greening technology investment.

**Case** **1.**
*The preorder quantity of the retailer is assumed to be less than the production volume of the manufacturer. Substitute $Q_{A1}^{*}=F^{-1}\left(\frac{w\;-\;c_{m}\;-\;te\left(1\;-\;\Delta e_{A}^{*}\right)}{w\;-\;v}\right)$ into Equation (3) when $q_{A}<Q_{A}^{*}$. Then, we will obtain the optimal preorder quantity of the retailer $q_{A1}^{*}$.*


**Proposition** **3.**
*There exists an advance purchase discount contract $\left(r,w,\;q_{A1}^{*},Q_{A1}^{*}\right)$. The optimal preorder quantity of the retailer is $q_{A1}^{*}=F^{-1}\left(\frac{w\;-\;\frac{w}{\left(1\;+\;r\right)}}{w\;-\;v}\right)$ (i.e., the optimal prepayment of the retailer is $D_{A1}^{*}=\frac{wq_{A1}^{*}}{\left(1\;+\;r\right)}\;$). The manufacturer’s optimal production volume and optimal carbon emission reduction rate are $Q_{A1}^{*}=F^{-1}\left(\frac{w\;-\;c_{m}\;-\;te\left(1\;-\;\Delta e_{A}^{*}\right)}{w\;-\;v}\right)$ and $\Delta e_{A1}^{*}=\frac{teQ_{A1}^{*}}{\xi }$. Furthermore, the discount factor satisfies the following conditions:$\;0<r<r_{1}$, $r_{1}=\frac{w}{c_{m}\;+\;te\left(1\;-\;\Delta e_{A}^{*}\right)}\;-\;1$.*


Proof. See Appendix A. 

**Corollary** **1.**
*The manufacturer’s optimal production volume and optimal carbon emission reduction rate with the APD contract $\left(r,w,\;q_{A1}^{*},Q_{A1}^{*}\right)\;$ are equal to those under the decentralized system, i.e., $Q_{A1}^{*}=Q^{*}$, $\Delta e_{A1}^{*}=\Delta e^{*}$.*


$\frac{w}{\left(1\;+\;r\right)}>c_{m}\;+\;te\left(1\;-\;\Delta e_{A}^{*}\right)$, i.e.,$\;0<r<r_{1}$. It represents that the manufacturer has non-negative profit even if she offers discounts. Thus, the contract $\left(r,w,\;q_{A1}^{*},Q_{A1}^{*}\right)\;$ is just a guarantee that the manufacturer will not lose. However, the retailer’s prepayment is not enough to change the manufacturer’s production volume and investment of greening technology $(Q_{A1}^{*}=Q^{*}$, $\Delta e_{A1}^{*}=\Delta e^{*})$, under the contract $\left(r,w,\;q_{A1}^{*},Q_{A1}^{*}\right)$. The discount factor r only affects the optimal preorder quantity $q_{A1}^{*}$ and has no influence on the optimal production volume $Q_{A1}^{*}$. If actual demand is less than the production volume of the manufacturer (scenario 1 and scenario 2 in Table 2), she still needs to bear the inventory risk. If actual demand is less than the preorder quantity of the retailer (scenario 1 in Table 2), he takes inventory risk. However, if actual demand is higher than the production volume of the manufacturer (scenario 3 in Table 2), the retailer will not be able to meet demand.

**Case** **2.**
*The preorder quantity of the retailer is assumed to be greater than the production volume of the manufacturer. Substitute $Q_{A2}^{*}=q_{A}$ into Equation (3) when $q_{A}\geq Q_{A}^{*}$. Then, we will obtain the retailer’s optimal decision $q_{A2}^{*}$.*


**Proposition** **4.**
*There exists an advance purchase discount contract $\left(r,w,\;q_{A2}^{*},Q_{A2}^{*}\right)$. The optimal preorder quantity of the retailer is $\;q_{A2}^{*}=F^{-1}\left(\frac{p\;-\;\frac{w}{1\;+\;r}}{p\;-\;v}\right)\;$ (i.e., the optimal prepayment of the retailer is $D_{A2}^{*}=\frac{wq_{A2}^{*}}{\left(1\;+\;r\right)}\;$). The manufacturer’s optimal production volume is $\;Q_{A2}^{*}=q_{A2}^{*}=F^{-1}\left(\frac{p\;-\;\frac{w}{1\;+\;r}}{p\;-\;v}\right)$ and her optimal carbon emission reduction rate is $\Delta e_{A2}^{*}=\frac{teQ_{A2}^{*}}{\xi }$. Furthermore, the discount factor satisfies the following conditions: $\;r_{0}<r<r_{1}$, $r_{0}=\frac{w}{p\left[1\;-\;F\left(Q_{A}^{*}\right)\right]\;+\;vF\left(Q_{A}^{*}\right)}\;-\;1$.*


Proof. See Appendix A. 

There exists a critical value of the discount factor $r_{0}$. $\left(r_{0},r_{1}\right)$ reflects the manufacturer’s strategy of the APD contract. As the retailer’s retail price decreases, the manufacturer will have a high floor of the discount factor, i.e.,$\;\frac{\partial r_{0}}{\partial p}<0$. The range of the discount factor is shown in Figure 3 as the carbon emission reduction rate $\Delta e_{A}^{*}$ increases. Further, Figure 3 shows that the manufacturer’s discount factor r increases with $\Delta e_{A}^{*}$.

**Corollary** **2.**
*The optimal preorder quantity of the retailer increases with the discount factor while it decreases with wholesale price, i.e., $\frac{\partial q_{A2}^{*}}{\partial r}>0$, $\frac{\partial q_{A2}^{*}}{\partial w}<0$.*


Corollary 2 suggests that the discount factor r affects the optimal preorder quantity $q_{A2}^{*}$. The retailer will increase his preorder quantity when the discount factor increases, in practice, because a high discount factor results in a low order cost for the retailer. However, the higher the wholesale price, the higher the second-order cost of the retailer. Therefore, the retailer prefers to order more products in advance.

**Corollary** **3.**
*The manufacturer’s optimal production volume and optimal carbon emission reduction rate with the APD contract $\left(r,w,\;q_{A2}^{*},Q_{A2}^{*}\right)\;$ are greater than those under the decentralized system, i.e., $Q_{A2}^{*}>Q^{*}$, $\Delta e_{A2}^{*}>\Delta e^{*}$.*


When the discount factor is higher than a certain threshold $r_{0}$, the retailer’s prepayment will make the manufacturer’s capacity and investment of greening technology higher than without the prepayment, i.e., $Q_{A2}^{*}>Q^{*}$*,*
$\Delta e_{A2}^{*}>\Delta e^{*}$. At this moment, the manufacturer provides a sufficient discount to reduce the retailer’s cost of ordering, which attracts the retailer to purchase more in advance, i.e., $q_{A2}^{*}>q_{A1}^{*}$. More payment is transferred from the cash on delivery at $T_{1}$ to prepay at $T_{0}$. Meanwhile, the manufacturer has more capital to invest in production and greening technology before the selling season. That means the manufacturer’s cash flow is improved by the prepayment.

For avoiding inventory risks, the manufacturer determines production volume according to the preorder quantity when $q_{A}>Q_{A}^{*}$, that is, $Q_{A2}^{*}=q_{A2}^{*}$. If actual demand is lower than the preorder quantity of the retailer (scenario 1 in Table 3), he bears inventory risk. If actual demand is higher than the preorder quantity of the retailer (scenario 2 in Table 3), stock shortage risk is also borne by her. Compared with the decentralized system, inventory risk is transferred from manufacturer to retailer. Therefore, from the perspective of avoiding inventory risk, the APD contract benefits the manufacturer.

**Proposition** **5.**
*$Q_{A2}^{*}=q_{A2}^{*}=Q_{c}^{*}$, $\Delta e_{A2}^{*}=\Delta e_{c}^{*}$*if and only if*$r^{*}=\frac{w\;-\;p\left[1\;-\;F\left(Q_{c}^{*}\right)\right]\;-\;vF\left(Q_{c}^{*}\right)}{p\left[1\;-\;F\left(Q_{c}^{*}\right)\right]\;+\;vF\left(Q_{c}^{*}\right)}$.*


Proof. See Appendix A. 

From Proposition 5, we know that if $r^{*}=\frac{w\;-\;p\left[1\;-\;F\left(Q_{c}^{*}\right)\right]\;-\;vF\left(Q_{c}^{*}\right)}{p\left[1\;-\;F\left(Q_{c}^{*}\right)\right]\;+\;vF\left(Q_{c}^{*}\right)}$, both the preorder quantity of the retailer and the production volume of the manufacturer are equal to the optimal production volume under the centralized system. Meanwhile, the manufacturer’s optimal carbon emission reduction rate with the contract equals that under the centralized system. Then, supply chain coordination can be reached with the APD contract $\left(r,w,\;q_{A2}^{*},Q_{A2}^{*}\right)$. The discount factor and the wholesale price are positively correlated under coordination, i.e.,$\;\frac{\partial r^{*}}{\partial w}>0$. If the wholesale price increases, the discount factor also increases. In other words, the manufacturer’s net profit on preorders does not increase. Then, the manufacturer only can increase her income by increasing the preorder quantity (i.e., $\;q_{A2}^{*}=Q_{c}^{*}$).

**Proposition** **6.**
*The optimal carbon emission reduction rate of the manufacturer is increasing in the carbon emission tax, i.e.,$\;\frac{\partial \Delta e_{A2}^{*}}{\partial t}>0$. The total profit of the manufacturer $\Pi_{m}$ is convex in the carbon emission tax t.*


Proof. See Appendix A. 

Substituting $\Delta e_{A2}^{*}=\frac{teQ_{A2}^{*}}{\xi }$ into Equation (4), we have $\frac{\partial^{2}\Pi_{m}}{\partial t^{2}}>0$. When $t\in (-\infty ,\frac{\xi }{eQ_{A2}^{*}})$, the profit of the manufacturer is a decreasing function. When $t\in \left(\frac{\xi }{eQ_{A2}^{*}},+\infty \right)$, the profit of the manufacturer is an increasing function. Therefore, when the carbon emission tax increases, total profit of the manufacturer first decreases and then increases as shown in Figure 4 and Figure 5.

### 5.2. The Option Contract

Under the option contract, firstly, the retailer optimizes option order quantity $q_{O}^{*}$. Then, the manufacturer optimizes production volume $Q_{O}^{*}$ and carbon emission reduction rate $\Delta e_{O}^{*}$. The cash flow of the supply chain with the option contract is as follows (see Figure 6):

$T_{0}$: Retailer pays the option fee $c_{o}q_{O}$ in advance.

$T_{1}$: $Q_{O}\;-\;\int_{0}^{Q_{O}}F\left(x\right)dx$ represents the retailer’s expected sales and his total sales revenue would be $p\left[Q_{O}\;-\;\int_{0}^{Q_{O}}F\left(x\right)dx\right]$. $c_{e}\left[q_{O}\;-\;\int_{0}^{q_{O}}F\left(x\right)dx\right]\;$ is the retailer’s cost of exercising options and his expected transfer payment for the second order is $w\left[Q_{O}\;-\;\int_{q_{O}}^{Q_{O}}F\left(x\right)dx\;-\;q_{O}\right]$.

$T_{2}$: $v[Q_{O}\;-\;(Q_{O}\;-\;\int_{o}^{Q_{O}}F\left(x\right)dx)]=v\int_{o}^{Q_{O}}F\left(x\right)dx$ is the manufacturer’s total expected salvage value. Then, the manufacturer pays carbon emission tax $te\left(1\;-\;\Delta e_{O}\right)Q_{O}$.

Then, the total profit of the retailer will be
(6)$$\begin{array}{c} \Pi_{r}\left(q_{O}\right)=p\left[Q_{O}\;-\;\int_{0}^{Q_{O}}F\left(x\right)dx\right]\;-\;c_{e}\left[q_{O}\;-\;\int_{0}^{q_{O}}F\left(x\right)dx\right]\\ -c_{0}q_{O}\;-\;w\left[Q_{O}\;-\;\int_{q_{O}}^{Q_{O}}F\left(x\right)dx\;-\;q_{O}\right]. \end{array}$$

In Equation (6), the first term is sales revenue, the second and third terms are exercising and initial ordering cost of options and the last term is the cost of the second order.

The total profit of the manufacturer will be
(7)$$\begin{array}{c} \Pi_{m}\left(Q_{O},\Delta e_{O}\right)=c_{0}q_{O}\;+\;c_{e}\;[q_{O}\;-\;\int_{0}^{q_{O}}F\left(x\right)dx]\;+\;w\left[Q_{O}\;-\;\int_{q_{O}}^{Q_{O}}F\left(x\right)dx\;-\;q_{O}\right]\\ \;+\;v\int_{o}^{Q_{O}}F\left(x\right)dx\;-\;\left(c_{m}\;+\;te\left(1\;-\;\Delta e_{O}\right)\right)Q_{O}\;-\;\frac{1}{2}\xi {\left(\Delta e_{O}\right)}^{2}\;, \end{array}$$
(8)$$S.t.q_{O}\leq Q_{O}.$$

In Equation (7), the first, second and third terms are revenues from retailer purchases and exercises of options and second-order purchases, respectively. The fourth term is total salvage value. The fifth term is total cost of production and carbon emission tax. The last term is cost of greening technology investment. The manufacturer needs to ensure the option order quantity. Then, her production volume satisfies the constraint $Q_{O}\geq q_{O}$.

Two cases are considered: (i) the option order quantity of the retailer is lower than the production volume of the manufacturer, (ii) vice versa. Using the Lagrangian relaxation approach, the manufacturer’s optimal production volume is $Q_{O1}^{*}=Q_{O}^{*}=F^{-1}\left(\frac{w\;-\;c_{m}\;-\;te\left(1\;-\;\Delta e_{O}^{*}\right)\;}{w\;-\;v}\right)$ and her optimal carbon emission reduction rate is $\Delta e_{O1}^{*}=\Delta e_{O}^{*}=\frac{teQ_{O}^{*}}{\xi }$ when $q_{O}<Q_{O}^{*}$. However, the manufacturer’s optimal production volume is $Q_{O2}^{*}=q_{O}$ and her optimal carbon emission reduction rate is $\;\Delta e_{O2}^{*}=\frac{teq_{O}}{\xi }$ when $\;q_{O}\geq Q_{O}^{*}$. That means the manufacturer’s optimal production volume with the option contract is $Q_{O}^{*}=max\left(q_{O},Q_{O}^{*}\right)$. This result is similar to that of [49], although their model does not consider the carbon emission tax and greening technology investment.

**Case** **1.**
*The option order quantity of the retailer is assumed to be less than the production volume of the manufacturer. We obtain the retailer’s optimal option order quantity decision $q_{O1}^{*}$ by substituting $\;Q_{O1}^{*}=F^{-1}\left(\frac{w\;-\;c_{m}\;-\;te\left(1\;-\;\Delta e_{O}^{*}\right)\;}{w\;-\;v}\right)$ into Equation (6) when $q_{O}<Q_{O}^{*}$.*


**Proposition** **7.**
*There exists an option contract $\left(c_{0},c_{e},\;q_{O1}^{*},Q_{O1}^{*}\right)$. The retailer’s optimal option order quantity is $q_{O1}^{*}=F^{-1}\left(\frac{w\;-\;c_{e}\;-\;c_{0}\;}{w\;-\;c_{e}}\right)$. The manufacturer’s optimal production volume is $Q_{O1}^{*}=F^{-1}\left(\frac{w\;-\;c_{m}\;-\;te\left(1\;-\;\Delta e_{O}^{*}\right)\;}{w\;-\;v}\right)$ and her optimal carbon emission reduction rate is $\Delta e_{O1}^{*}=\frac{teQ_{O1}^{*}}{\xi }$. Furthermore, the option price satisfies the following conditions:$\;\left(w\;-\;c_{e}\right)\left[1\;-\;F\left(Q_{O}^{*}\right)\right]<c_{0}$.*


Proof. See Appendix A. 

**Corollary** **4.**
*The manufacturer’s optimal production volume and optimal carbon emission reduction rate with the option contract $\left(c_{0},c_{e},\;q_{O1}^{*},Q_{O1}^{*}\right)\;$ are equal to those under the decentralized system, i.e., $Q_{O1}^{*}=Q^{*}$, $\Delta e_{O1}^{*}=\Delta e^{*}$.*


If the option price can rise without a cap, the option contract $\left(c_{0},c_{e},\;q_{O1}^{*},Q_{O1}^{*}\right)$ cannot change the manufacturer’s production volume and investment of greening technology, i.e., $\;Q_{O1}^{*}=Q^{*}$*,*
$\Delta e_{O1}^{*}=\Delta e^{*}$. Similar to Proposition 3, the strike price $c_{e}$ and the option price $c_{0}$ only affect the optimal option order quantity $q_{O1}^{*}$ and have no influence on the optimal production volume $Q_{O1}^{*}$. Regardless of whether actual demand is less or more than the option order quantity of the retailer (scenario 1 and scenario 2 in Table 4), the manufacturer always needs to bear the inventory risk. When actual demand is higher than the manufacturer’s output (scenario 3 in Table 4), the retailer will not be able to meet demand.

**Case** **2.**
*The option order quantity of the retailer is assumed to be greater than the production volume of the manufacturer. We will obtain the retailer’s optimal decision $q_{O2}^{*}$ by substituting $\;Q_{O2}^{*}=q_{O}$ into Equation (6) when $q_{O}\geq Q_{O}^{*}$.*


**Proposition** **8.**
*There exists an option contract $\left(c_{0},c_{e},\;q_{O2}^{*},Q_{O2}^{*}\right)$. The retailer’s optimal option order quantity is $q_{O2}^{*}=F^{-1}\left(\frac{p\;-\;c_{e}\;-\;c_{0}\;}{p\;-\;c_{e}}\right)$. The manufacturer’s optimal production volume is $Q_{O2}^{*}=q_{O2}^{*}=F^{-1}\left(\frac{p\;-\;c_{e}\;-\;c_{0}\;}{p\;-\;c_{e}}\right)$ and her optimal carbon emission reduction rate is $\Delta e_{O2}^{*}=\frac{teQ_{O2}^{*}}{\xi }$. Furthermore, the option price satisfies the following conditions: $\;c_{01}<c_{0}<c_{02}$, $c_{01}=c_{m}\;+\;te\left(1\;-\;\Delta e_{O}^{*}\right)\;-\;c_{e}$, $c_{02}=\left(p\;-\;c_{e}\right)\left[1\;-\;F\left(Q_{O}^{*}\right)\right]$.*


Proof. See Appendix A. 

$c_{01}$ represents that the manufacturer has non-negative profit with the option contract.$\;c_{02}$ is a critical value of option price. $\left(c_{01},c_{02}\right)$ reflects the manufacturer’s strategy of the option contract. If retail price is high, the manufacturer will have a high cap of the option price, i.e., $\;\frac{\partial c_{02}}{\partial p}>0$. If strike price is high, the manufacturer will have a low cap of the option price, i.e., $\;\frac{\partial c_{02}}{\partial c_{e}}<0$. Further, Figure 7 shows that the option price $c_{0}$ is decreasing in the carbon emission reduction rate $\Delta e_{O}^{*}$.

**Corollary** **5.**
*The optimal option order quantity of the retailer is decreasing in option price and strike price, i.e., $\frac{\partial q_{O2}^{*}}{\partial c_{e}}<0$, $\frac{\partial q_{O2}^{*}}{\partial c_{0}}<0$.*


Corollary 5 demonstrates that the retailer prefers an option contract with a low option price and a low strike price because it means low risk and low ordering cost. Different from the APD contract $\;\left(r,w,\;q_{A2}^{*},Q_{A2}^{*}\right)$, the optimal option order quantity of the retailer $q_{O2}^{*}\;$ is independent of the wholesale price w. Since $c_{0}\;+\;c_{e}<w$, the manufacturer can guarantee that the price is favorable to attract the retailer mainly by the option mechanism instead of the instantaneous purchase mechanism [49]. Hence, the wholesale price does not affect the optimal option order quantity of the retailer.

**Corollary** **6.**
*The manufacturer’s optimal production volume and optimal carbon emission reduction rate with the option contract $\left(c_{0},c_{e},\;q_{O2}^{*},Q_{O2}^{*}\right)$ are greater than those under the decentralized system, i.e., $Q_{O2}^{*}>Q^{*}$, $\Delta e_{O2}^{*}>\Delta e^{*}$.*


When the option price is lower than a certain threshold $c_{02}$, the retailer’s option fee will make the manufacturer’s production capacity and investment of greening technology higher than without a contract, i.e., $Q_{A2}^{*}>Q^{*}$*,*
$\Delta e_{A2}^{*}>\Delta e^{*}$. That means the manufacturer’s cash flow is improved by the option fee. The low option price $\left(c_{01}<c_{0}<c_{02}\right)\;$ attracts the retailer to order more options in advance, i.e., $q_{O2}^{*}>q_{O1}^{*}$. Then, the manufacturer has more capital to invest in production and greening technology.

Since $Q_{O2}^{*}=q_{O2}^{*}$*,* inventory risk is entirely borne by the manufacturer when actual demand is less than the option order quantity of the retailer (scenario 1 in Table 5). Therefore, from the perspective of avoiding inventory risk, the option contract benefits the retailer. However, due to the limitation of the manufacturer’s production, the retailer cannot replenish any more when actual demand is higher than the production volume of the manufacturer (scenario 2 in Table 5). Compared with the option contract $\left(c_{0},c_{e},\;q_{O1}^{*},Q_{O1}^{*}\right)$, the option contract $\left(c_{0},c_{e},\;q_{O2}^{*},Q_{O2}^{*}\right)\;$ reduces the manufacturer’s inventory risk when actual demand is greater than the option order quantity of the retailer and less than the production volume of the manufacturer (scenario 2 in Table 4).

**Proposition** **9.**
*$\;q_{O2}^{*}=Q_{O2}^{*}=Q_{c}^{*}$, $\Delta e_{O2}^{*}=\Delta e_{c}^{*}$*if and only if*$c_{0}^{*}=\left(p\;-\;c_{e}\right)\left[1\;-\;F\left(Q_{c}^{*}\right)\right]$.*


Proof. See Appendix A. 

From Proposition 9, we know that when $c_{0}^{*}=\left(p\;-\;c_{e}\right)\left[1\;-\;F\left(Q_{c}^{*}\right)\right]$, both the option order quantity of the retailer and the production volume of the manufacturer are equal to the optimal production volume under the centralized supply chain. Meanwhile, the manufacturer’s optimal carbon emission reduction rate with the contract equals that under the centralized system. That means supply chain coordination can be attained with the option contract $\left(c_{0},c_{e},\;q_{O2}^{*},Q_{O2}^{*}\right)$. There is a negative correlation between option price and strike price under coordination, i.e., $\;\frac{\partial c_{0}^{*}}{\partial c_{e}}<0$, since option price brings a guaranteed revenue for the manufacturer while strike price leads to uncertainty of future earnings (the retailer may not exercise options, or a second order may not happen). If the option price increases, the manufacturer obtains more payments for an option fee. However, the decline in strike price will cause a decline in the manufacturer’s potential future earnings. Therefore, the manufacturer could achieve a delicate balance with an appropriate option order quantity (i.e., $\;q_{O2}^{*}=Q_{c}^{*}$).

**Proposition** **10.**
*The optimal carbon emission reduction rate of the manufacturer is increasing in carbon emission tax, i.e., $\;\frac{\partial \Delta e_{O2}^{*}}{\partial t}>0$. The total profit of the manufacturer $\Pi_{m}$ is convex in the carbon emission tax t.*


Proof. See Appendix A. 

Substituting $\Delta e_{O2}^{*}=\frac{teQ_{O2}^{*}}{\xi }$ into Equation (7), we have $\frac{\partial^{2}\Pi_{m}}{\partial t^{2}}>0$. When $t\in \left(-\infty ,\frac{\xi }{eQ_{O2}^{*}}\right)$, the profit of the manufacturer is a decreasing function. When $t\in \left(\frac{\xi }{eQ_{O2}^{*}},+\infty \right)$, the profit of the manufacturer is an increasing function. Therefore, when carbon emission tax increases, total profit of the manufacturer first decreases and then increases as shown in Figure 4 and Figure 5.

### 5.3. The Prepayment-Based Option (PBO) Contract

Under the PBO contract, firstly, the retailer optimizes option order quantity $y_{O}^{*}$ and preorder quantity $\;y_{A}^{*}$. Then, the manufacturer optimizes production volume $Q_{M}^{*}$ and carbon emission reduction rate $\Delta e_{M}^{*}$. The cash flow of supply chain with the PBO contract is as follows (see Figure 8):

$T_{0}$: Retailer pays $\frac{y_{A}w}{1\;+\;r}\;+\;c_{0}y_{o}$ in advance for the preorder quantity and the option order quantity.

$T_{1}$: $Q_{M}\;-\;\int_{0}^{Q_{M}}F\left(x\right)dx$ represents the retailer’s expected sales and his total sales revenue would be $p\left[Q_{M}\;-\;\int_{0}^{Q_{M}}F\left(x\right)dx\right]$. The number of options to be exercised is $\;y_{o}\;+\;y_{A}\;-\;\int_{0}^{y_{o}+y_{A}}F\left(x\right)dx\;+\;\int_{0}^{y_{A}}F\left(x\right)dx\;-\;y_{A}=y_{o}\;-\;\int_{y_{A}}^{y_{o}+y_{A}}F\left(x\right)dx$. Meanwhile, the retailer’s transfer payment for exercising options is $\;c_{e}\left[y_{o}\;-\;\int_{y_{A}}^{y_{o}+y_{A}}F\left(x\right)dx\right]$. $Q_{M}\;-\;\int_{o}^{Q_{M}}F\left(x\right)dx\;+\;\int_{0}^{y_{o}+y_{A}}F\left(x\right)dx\;-\;\left(y_{o}\;+\;y_{A}\right)=Q_{M}\;-\;\int_{y_{o}+y_{A}}^{Q_{M}}F\left(x\right)dx\;-\;\left(y_{o}\;+\;y_{A}\right)\;$ is expected sales at wholesale price w of the manufacturer. Meanwhile, the transfer payment for the retailer’s second order is $\;w[Q_{M}\;-\;\int_{y_{o}+y_{A}}^{Q_{M}}F\left(x\right)dx\;-\;\left(y_{o}\;+\;y_{A}\right)]$.

$T_{2}$: The retailer’s expected inventory is $y_{A}\;-\;\left(y_{A}\;-\;\int_{0}^{y_{A}}F\left(x\right)dx\right)=\int_{0}^{y_{A}}F\left(x\right)dx$ while the manufacturer’s expected inventory is $Q_{M}\;-\;\left(Q_{M}\;-\;\int_{0}^{Q_{M}}F\left(x\right)dx\;+\;\int_{0}^{y_{A}}F\left(x\right)dx\right)=\int_{y_{A}}^{Q_{M}}F\left(x\right)dx$. The per-unit salvage value is v. Then, the manufacturer pays carbon emission tax $te\left(1\;-\;\Delta e_{M}\right)Q_{M}$.

Then, the retailer’s expected profit will be
(9)$$\begin{array}{c} \Pi_{r}\left(y_{o},y_{A}\right)=p\left[Q_{M}\;-\;\int_{0}^{Q_{M}}F\left(x\right)dx\right]\;-\;\frac{y_{A}w}{1\;+\;r}\;-\;c_{0}y_{o}\;-\;c_{e}\left[y_{o}\;-\;\int_{y_{A}}^{y_{o}+y_{A}}F\left(x\right)dx\right]\\ \;-\;w[Q_{M}\;-\;\int_{y_{o}+y_{A}}^{Q_{M}}F\left(x\right)dx\;-\;\left(y_{o}\;+\;y_{A}\right)]\;+\;v\int_{o}^{y_{A}}F\left(x\right)dx. \end{array}$$

In Equation (9), the first term is total sales revenue, the second term is prepayment for preorder quantity, the third term is cost of buying options, the fourth term denotes cost of exercising options, the fifth term is cost of a second order and the last term is total salvage value.

The manufacturer’s expected profit will be
(10)$$\begin{array}{c} \Pi_{m}\left(Q_{M},\Delta e_{M}\right)=\frac{y_{A}w}{1\;+\;r}\;+\;c_{0}y_{o}\;+\;c_{e}\;+\;w[Q_{M}\;-\;\int_{y_{o}+y_{A}}^{Q_{M}}F\left(x\right)dx\;-\;\left(y_{o}\;+\;y_{A}\right)]\\ \;+\;v\int_{y_{A}}^{Q_{M}}F\left(x\right)dx\;-\;\left(c_{m}\;+\;te\left(1\;-\;\Delta e_{M}\right)\right)Q_{M}\;-\;\frac{1}{2}\xi {\left(\Delta e_{M}\right)}^{2}\;, \end{array}$$
(11)$$S.t.\;y_{o}\;+\;y_{A}\leq Q_{M}\;.$$

In Equation (10), the first, second, third and fourth terms are revenue from preselling, options selling and exercising and second-selling, respectively. The fifth term is total salvage value. The sixth term is total cost of production and carbon emission tax. The last term is cost of greening technology investment. The manufacturer has to ensure the preorder and the option order quantity. Therefore, her production volume satisfies the constraint $Q_{M}\geq y_{o}\;+\;y_{A}$.

Two cases are considered: (i) the total order quantity of the retailer is lower than the production volume of the manufacturer, (ii) vice versa. Using the Lagrangian relaxation approach, the manufacturer’s optimal production volume is $\;Q_{M1}^{*}=Q_{M}^{*}=F^{-1}\left(\frac{w\;-\;c_{m}\;-\;te\left(1\;-\;\Delta e_{M}^{*}\right)}{w\;-\;v}\right)$ and her optimal carbon emission reduction rate is $\Delta e_{M1}^{*}=\Delta e_{M}^{*}=\frac{teQ_{M}^{*}}{\xi }$ when $y_{o}\;+\;y_{A}<Q_{M}^{*}$. However, the manufacturer’s optimal production volume is $Q_{M2}^{*}=y_{o}\;+\;y_{A}$ and her optimal carbon emission reduction rate is $\Delta e_{M2}^{*}=\frac{te\left(y_{o}\;+\;y_{A}\right)}{\xi }$ when $y_{o}\;+\;y_{A}\geq Q_{M}^{*}$. That means the manufacturer’s optimal production volume with the PBO contract is $Q_{M}^{*}=max\left(y_{o}\;+\;y_{A},Q_{M}^{*}\right)$.

**Case** **1.**
*The total order quantity of the retailer is assumed to be lower than the production volume of the manufacturer. Substituting $\;Q_{M1}^{*}=F^{-1}\left(\frac{w\;-\;c_{m}\;-\;te\left(1\;-\;\Delta e_{M}^{*}\right)}{w\;-\;v}\right)$ into Equation (9) when $y_{o}\;+\;y_{A}<Q_{M}^{*}$, we will obtain the retailer’s optimal preorder quantity $y_{A1}^{*}$ and his optimal option order quantity $y_{O1}^{*}$.*


**Proposition** **11.**
*There exists a prepayment-based option contract $\;(r,c_{0},c_{e},\;y_{A1}^{*},y_{O1}^{*},Q_{M1}^{*})$. The retailer’s optimal preorder quantity is $\;y_{A1}^{*}=F^{-1}\left(\frac{c_{e}\;+\;c_{0}\;-\;\frac{w}{\left(1\;+\;r\right)}}{c_{e}\;-\;v}\right)$ and his optimal option order quantity is $y_{O1}^{*}=F^{-1}\left(1\;-\;\frac{c_{0}}{w\;-\;c_{e}}\right)\;-\;F^{-1}\left(\frac{c_{e}\;+\;c_{0}\;-\;\frac{w}{\left(1\;+\;r\right)}}{c_{e}\;-\;v}\right)$. The manufacturer’s optimal production volume is $\;Q_{M1}^{*}=F^{-1}\left(\frac{w\;-\;c_{m}\;-\;te\left(1\;-\;\Delta e_{M}^{*}\right)\;}{w\;-\;v}\right)$ and her optimal carbon emission reduction rate is $\Delta e_{M1}^{*}=\frac{teQ_{M1}^{*}}{\xi }$. Furthermore, the option price satisfies the following conditions: $\;\left(w\;-\;c_{e}\right)\left[1\;-\;F\left(Q_{M}^{*}\right)\right]<c_{0}$.*


Proof. See Appendix A. 

**Corollary** **7.**
*The manufacturer’s optimal production volume and carbon emission reduction rate with the PBO contract $\;(r,c_{0},c_{e},\;y_{A1}^{*},y_{O1}^{*},Q_{M1}^{*})$ are equal to those under the decentralized system, i.e., $Q_{M1}^{*}=Q^{*}$, $\Delta e_{M1}^{*}=\Delta e^{*}$.*


Similar to Corollary 4, if the option price can rise without a cap, the PBO contract $\;(r,c_{0},c_{e},\;y_{A1}^{*},y_{O1}^{*},Q_{M1}^{*})$ cannot change the manufacturer’s production volume and investment of greening technology, i.e., $Q_{M1}^{*}=Q^{*}$*,*
$\Delta e_{M1}^{*}=\Delta e^{*}$. Whether actual demand is lower or higher than the total order quantity of the retailer (scenario 1, scenario 2 and scenario 3 in Table 6), the manufacturer always needs to bear inventory risk. The retailer just bears inventory risk when actual demand is less than his preorder quantity (scenario 1 in Table 6). However, when actual demand is higher than the production volume of the manufacturer, the retailer will not be able to meet demand (scenario 4 in Table 6).

**Case** **2.**
*The total order quantity of the retailer is assumed to be greater than the production volume of the manufacturer. Substitute $\;Q_{M2}^{*}=y_{o}\;+\;y_{A}$ into Equation (9) when $y_{o}\;+\;y_{A}\geq Q_{M}^{*}$. Then, we will obtain the retailer’s optimal preorder quantity $y_{A2}^{*}$ and his option order quantity $y_{O2}^{*}$.*


**Proposition** **12.**
*There exists a prepayment-based option contract $\;(r,c_{0},c_{e},\;y_{A2}^{*},y_{O2}^{*},Q_{M2}^{*})$. The retailer’s optimal preorder quantity is $\;y_{A2}^{*}=F^{-1}\left(\frac{c_{e}\;+\;c_{0}\;-\;\frac{w}{\left(1\;+\;r\right)}}{\left(c_{e}\;-\;v\right)}\right)$ and his optimal option order quantity is $y_{O2}^{*}=F^{-1}\left(1\;-\;\frac{c_{0}}{p\;-\;c_{e}}\right)\;-\;F^{-1}\left(\frac{c_{e}\;+\;c_{0}\;-\;\frac{w}{\left(1\;+\;r\right)}}{\left(c_{e}\;-\;v\right)}\right)$. The manufacturer’s optimal production volume is $Q_{M2}^{*}=y_{A2}^{*}\;+\;y_{O2}^{*}=F^{-1}\left(1\;-\;\frac{c_{0}}{p\;-\;c_{e}}\right)$ and her optimal carbon emission reduction rate is $\Delta e_{M2}^{*}=\frac{teQ_{M2}^{*}}{\xi }$. Furthermore, the option price satisfies the following conditions: $\;c_{03}<c_{0}<c_{04}$, $c_{03}=c_{m}\;+\;te\left(1\;-\;\Delta e_{M}^{*}\right)\;-\;c_{e}$
$c_{04}=\left(p\;-\;c_{e}\right)\left[1\;-\;F\left(Q_{M}^{*}\right)\right]$.*


Proof. See Appendix A. 

**Corollary** **8.**
*The manufacturer’s optimal production volume and optimal carbon emission reduction rate with the PBO contract $\;(r,c_{0},c_{e},\;y_{A2}^{*},y_{O2}^{*},Q_{M2}^{*})$ is greater than that under the decentralized system, i.e., $Q_{M2}^{*}>Q^{*}$, $\Delta e_{M2}^{*}>\Delta e^{*}$.*


Similar to the option contract $\left(c_{0},c_{e},\;q_{O2}^{*},Q_{O2}^{*}\right)$, $\left(c_{03},c_{04}\right)$ reflects the manufacturer’s strategy of the PBO contract. The manufacturer’s option price $c_{0}$ is decreasing in $\Delta e_{M}^{*}=\Delta e_{O}^{*}=\Delta e^{*}$. When the option price is lower than a certain threshold $c_{04}$, the optimal production volume and the optimal carbon emission reduction rate are higher than without a contract, i.e., $Q_{M2}^{*}>Q^{*}$*,*
$\Delta e_{M2}^{*}>\Delta e^{*}$. That means the manufacturer’s cash flow is improved by the option fee. The lower option price attracts the retailer to order more options in advance, i.e., $\;y_{O2}^{*}>y_{O1}^{*}$ and $\;y_{A2}^{*}\;+\;y_{O2}^{*}>y_{A1}^{*}\;+\;y_{O1}^{*}$. Then, the manufacturer has more capital to invest in production and greening technology. However, the discount factor does not affect the manufacturer’s strategy of the PBO contract. Moreover, different from the APD contract, the preorder quantity under the PBO contract does not affect the manufacturer’s production volume and investment of greening technology, i.e., $\;y_{A2}^{*}=y_{A1}^{*}$.

If actual demand is lower than the preorder quantity of the retailer (scenario 1 in Table 7), the inventory risk is shared by manufacturer and retailer. The salvage value of the retailer is $v\left(\;y_{A2}^{*}\;-\;x\right)$ while the salvage value of the manufacturer is $v\left(Q_{M2}^{*}\;-\;y_{A2}^{*}\right)=vy_{O2}^{*}$. However, inventory risk is entirely borne by the manufacturer, under the single option contract. The retailer entirely bears inventory risk under the APD contract. Therefore, compared with the APD contract and the option contract, the PBO contract $\;(r,c_{0},c_{e},\;y_{A2}^{*},y_{O2}^{*},Q_{M2}^{*})$ generates intermediate allocations of inventory risk between manufacturer and retailer.

Compared with the decentralized system, inventory risk is fully borne by the manufacturer when actual demand is higher than the retailer’s preorder quantity and lower than the manufacturer’s production volume (scenario 2 in Table 7). However, the manufacturer also has the salvage value and option fee $\left(v\left(Q_{M2}^{*}\;-\;x\right)\;+\;c_{0}y_{O2}^{*}\right)$.

**Proposition** **13.**
*$\;\;y_{A2}^{*}\;+\;y_{O2}^{*}=Q_{M2}^{*}=Q_{c}^{*}$, $\Delta e_{M2}^{*}=\Delta e_{c}^{*}$*if and only if*$c_{0}^{*}=\left(p\;-\;c_{e}\right)\left[1\;-\;F\left(Q_{c}^{*}\right)\right]$.*


Proof. See Appendix A. 

From Proposition 13, we know that when $c_{0}^{*}=\left(p\;-\;c_{e}\right)\left[1\;-\;F\left(Q_{c}^{*}\right)\right]$, both the total order quantity of the retailer and the production volume of the manufacturer equal the optimal production volume under the centralized system. Meanwhile, the manufacturer’s optimal carbon emission reduction rate with a contract is equal to that under the centralized system. That means supply chain coordination could be reached with the PBO contract $\left(r,c_{0},c_{e},\;y_{A2}^{*},y_{O2}^{*},Q_{M2}^{*}\right)$. Similar to Proposition 9, there is a negative relationship between the option price and the strike price under coordination, i.e., $\;\frac{\partial c_{0}^{*}}{\partial c_{e}}<0$.

**Proposition** **14.**
*The optimal carbon emission reduction rate of the manufacturer is increasing in carbon emission tax, i.e., $\;\frac{\partial \Delta e_{M2}^{*}}{\partial t}>0$. The total profit of the manufacturer $\Pi_{m}$ is convex in the carbon emission tax t.*


Proof. See Appendix A. 

Substituting $\Delta e_{M2}^{*}=\frac{teQ_{M2}^{*}}{\xi }$ into Equation (10), we have $\frac{\partial^{2}\Pi_{m}}{\partial t^{2}}>0$. When $t\in \left(-\infty ,\frac{\xi }{eQ_{M2}^{*}}\right)$, the profit of the manufacturer is a decreasing function. When $t\in \left(\frac{\xi }{eQ_{M2}^{*}},+\infty \right)$, the profit of the manufacturer is an increasing function. Therefore, when carbon emission tax increases, the total profit of the manufacturer first decreases and then increases, as shown in Figure 4 and Figure 5. Compared with the APD contract $\left(w,r,\;q_{A2}^{*},Q_{A2}^{*}\right)$, the manufacturer’s total profit with the PBO contract $\left(r,c_{0},c_{e},\;y_{A2}^{*},y_{O2}^{*},Q_{M2}^{*}\right)$ is first lower as $t\in \left(\frac{\xi }{eQ_{M2}^{*}},+\infty \right)$ and then higher as $t\in \left(\frac{\xi }{eQ_{A2}^{*}},+\infty \right)$.

## 6. Discussion of Results

In case 1, the retailer earns a negative profit under coordination. Therefore, we are not interested in case 1. In case 2, some interesting results are found in our study. We summarize some implications of these conclusions for supply chain managers.

Firstly, for improving cash flow, the APD contract, the option contract and the PBO contract can help improve cash flow, expand production and enhance the manufacturer’s investment in greening technology.

From Corollaries 3, 6 and 8, we can see that $Q_{A2}^{*}/Q_{O2}^{*}/Q_{M2}^{*}>Q^{*},\;\Delta e_{A2}^{*}/\Delta e_{O2}^{*}/\Delta e_{M2}^{*}>\Delta e^{*}$. The supply chain can optimize capacity and carbon emission reduction, under contract. We considered preselling (or option selling) as an additional financing source for the manufacturer. The sufficient cash flow of the manufacturer increases her production volume and greening technology investment. From Propositions 4, 8 and 12, we found the threshold that would guarantee the interests of all parties under each contract. For example, when the discount factor is in a certain range ($r_{0},r_{1}$), the retailer can obtain enough discounts from prepayment while the manufacturer can also make it profitable. Then, the retailer will prepay more prepayment for more preorder quantity, under the APD contract. There are two ranges for option prices, $c_{0}\in \left(c_{01},c_{02}\right)$ in the option contract and $c_{0}\in \left(c_{03},c_{04}\right)$ in the PBO contract. If the option price is too low, the cost of the retailer exercising the option will be very low, and demand risk will be transferred to the manufacturer. If the option price is too high, it will affect the willingness of the retailer to buy options. When the option price is within a certain range, ($c_{01},c_{02}$) or $\left(c_{03},c_{04}\right)$, the retailer will buy more options in advance, under either the option contract or PBO contract. Therefore, the manufacturer has more capital to invest in production and greening technology before the selling season.

Secondly, from the perspective of avoiding inventory risk, the APD contract benefits the manufacturer while the option contract benefits the retailer. However, the PBO contract generates intermediate allocations of inventory risk between manufacturer and retailer.

According to Propositions 5, 9 and 13, it can be seen that the supply chain coordination can be reached with these contracts. Under coordination, the manufacturer’s optimal capacity has expanded as Proposition 15. Meanwhile, stochastic demand is considered in our model. Then, these contracts change the inventory risk-sharing between retailer and manufacturer. The retailer never has extra products under the option contract. The number of options exercised is equal to the realized demand. However, the manufacturer never has surplus inventory under the APD contract. Her output equals the preorder quantity. Thus, from the perspective of avoiding inventory risk, the option contract benefits the retailer while the APD contract benefits the manufacturer. Under the PBO contract, the manufacturer bears market risk of the option order quantity while the retailer bears market risk of the preorder quantity. The retailer can maximize their own profit by predicting market demand information and deciding whether to exercise options. The manufacturer uses discounts to attract the retailer to increase preorders and payment in advance, reducing the pressure on capital and hedging risk. The retailer transfers market risk to the manufacturer by buying options, while the manufacturer protects herself by prepaying. The integration of the two contracts avoids the excessive risk of one party. Therefore, the PBO contract generates intermediate allocations of inventory risk between manufacturer and retailer. More importantly, the integration of the two contracts not only improves the capacity and green performance of the supply chain, but also realizes risk-sharing.

Lastly, the manufacturer’s optimal carbon emission reduction rate and the carbon emission tax are positively related while the retailer’s optimal order decision is not influenced by the carbon emission tax. From Propositions 6, 10 and 14, we can see that $\;\frac{\partial \Delta e_{A2}^{*}}{\partial t}>0$, $\frac{\partial \Delta e_{O2}^{*}}{\partial t}>0$, $\frac{\partial \Delta e_{M2}^{*}}{\partial t}>0$. Under each contract, the retailer’s optimal order decision is not influenced by the carbon emission tax while the optimal carbon emission reduction rate of the manufacturer and the carbon emission tax are positively related. More importantly, when the carbon emission tax increases, the total profit of the manufacturer first decreases and then increases for each contract. The carbon emission tax regulation does not set a cap on total emissions for enterprises, but guides them to optimize production technology through tax, so as to achieve the goal of carbon emission reduction. In addition, a fixed carbon tax rate helps the enterprises avoid the risk of emission reduction caused by cost fluctuations.

## 7. Conclusions

We investigate the green supply chain coordination problem with contracts in a two-echelon supply chain, consisting of one retailer and one manufacturer, under carbon emission tax regulation by considering greening technology investment and stochastic demand.

We firstly derive the optimal operation and greening decisions for both the decentralized system and the centralized system, including the production volume and greening technology investment. Then, we obtain the optimal decisions under each contract (i.e., the APD contract, the option contract and the PBO contract). Two cases are considered for the retailer’s strategies: (i) the retailer’s order quantity (i.e., the preorder quantity under the APD contract, the option order quantity under the option contract, the total order quantity under the PBO contract) is assumed to be less than the manufacturer’s production volume, (ii) vice versa. We derive the supply chain coordination conditions under each contract. Meanwhile, we discuss the cash flow, the inventory risk allocation and the impacts of carbon emission tax under each contract. For improving cash flow, preselling (or option selling) as a means of supporting the manufacturer with sufficient cash flow will help expand production and invest in greening technology. For avoiding inventory risk, the APD contract benefits the manufacturer while the option contract benefits the retailer. However, the PBO contract generates intermediate allocations of inventory risk between manufacturer and retailer. It can be observed that the retailer’s optimal order decision is not influenced by the carbon emission tax while the optimal carbon emission reduction rate of the manufacturer and the carbon emission tax are positively related.

Future research will be conducted as follows. Firstly, only one manufacturer and one retailer case is considered in this study. The problem becomes complicated if more than one manufacturer or retailer are involved in the supply chain. It is another challenge to apply current existing contracts or design new contracts to coordinate the supply chain. Secondly, further study could consider the supply chain with risk-averse supply chain agents. Many decision makers are risk-averse in the finance and economics literature. Therefore, it is necessary to design contracts that achieve the coordination of this type of emission-dependent supply chain.

## Figures and Tables

**Figure 1 ijerph-19-09232-f001:**
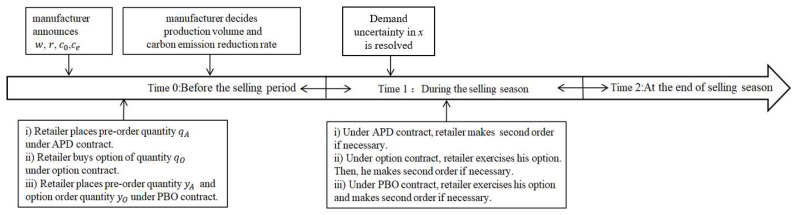
Sequence of events under contracts.

**Figure 2 ijerph-19-09232-f002:**
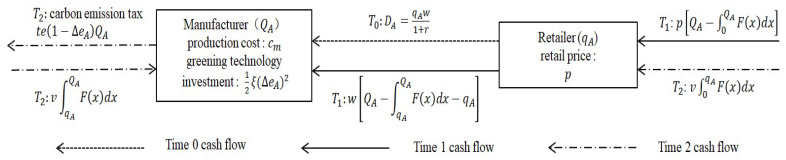
The cash flow of supply chain with the APD contract.

**Figure 3 ijerph-19-09232-f003:**
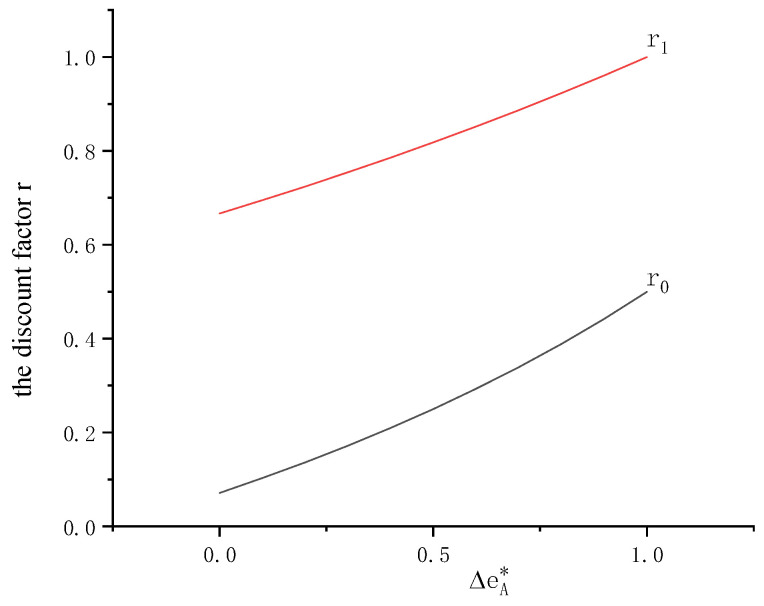
Range of the discount factor (*p* = 20, *w* = 10, *v* = 4, *t* = 0.5, *e* = 2).

**Figure 4 ijerph-19-09232-f004:**
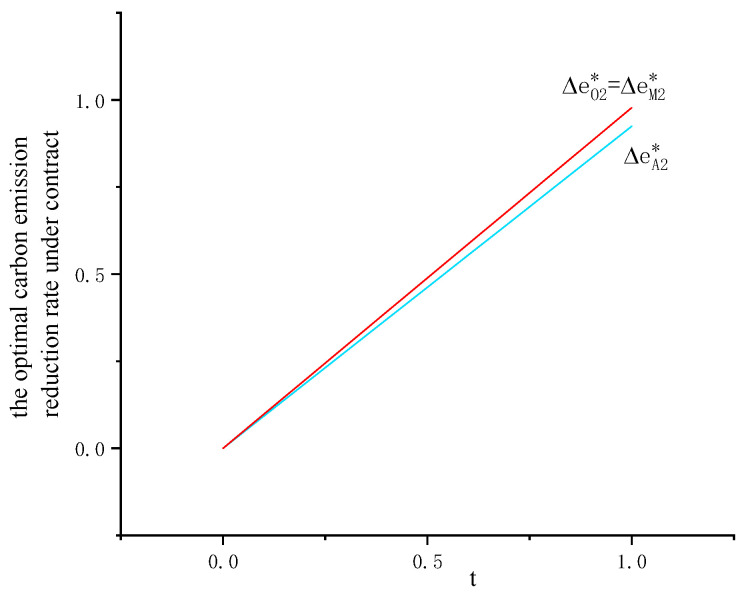
Influence of carbon emission tax on the carbon emission reduction rate under contracts (*r* = 0.3, *c*_0_ = 1.5).

**Figure 5 ijerph-19-09232-f005:**
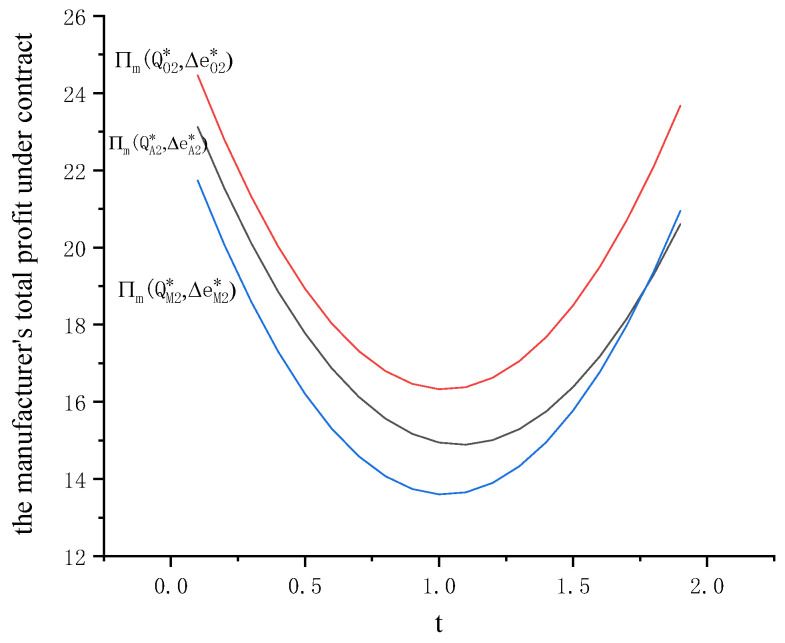
Influence of carbon emission tax on the total profit of manufacturer under contracts.

**Figure 6 ijerph-19-09232-f006:**
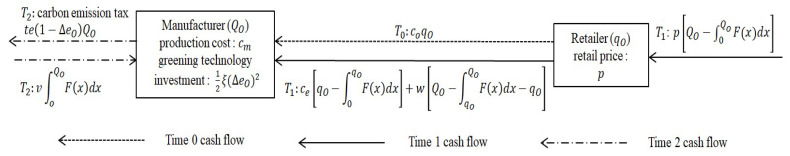
The cash flow of supply chain with the option contract.

**Figure 7 ijerph-19-09232-f007:**
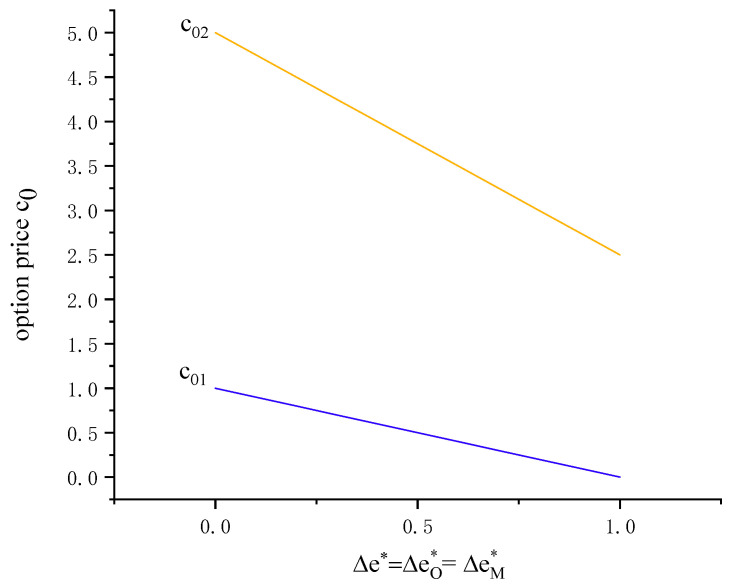
Range of option price (*p* = 20, *w* = 10, *v* = 4, *t* = 0.5, *e* = 2, *c_e_* = 5).

**Figure 8 ijerph-19-09232-f008:**
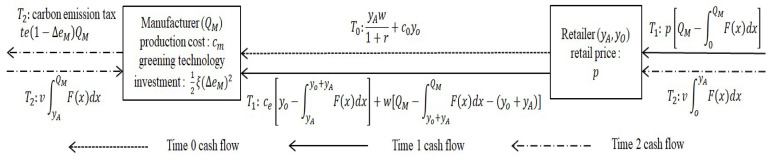
The cash flow of supply chain with PBO contract.

**Table 2 ijerph-19-09232-t002:** Risk-sharing of partners under APD contract $\left(r,w,\;q_{A1}^{*},Q_{A1}^{*}\right)$.

Actual Demand	Manufacturer	Retailer
$x<q_{A1}^{*}<Q_{A1}^{*}$	Leftover	Leftover
$q_{A1}^{*}<x<Q_{A1}^{*}$	Leftover	N/A
$q_{A1}^{*}<Q_{A1}^{*}<x$	N/A	Stockout

**Table 3 ijerph-19-09232-t003:** Risk-sharing of partners under the APD contract $\left(r,w,\;q_{A2}^{*},Q_{A2}^{*}\right)$.

Actual Demand	Manufacturer	Retailer
$x<q_{A2}^{*}=Q_{A2}^{*}$	N/A	Leftover
$q_{A2}^{*}=Q_{A2}^{*}<x$	N/A	Stockout

**Table 4 ijerph-19-09232-t004:** Risk-sharing of partners under the option contract $\left(c_{0},c_{e},\;q_{O1}^{*},Q_{O1}^{*}\right)$.

Actual Demand	Manufacturer	Retailer
$x<q_{O1}^{*}<Q_{O1}^{*}$	Leftover	N/A
$q_{O1}^{*}<x<Q_{O1}^{*}$	Leftover	N/A
$q_{O1}^{*}<Q_{O1}^{*}<x$	N/A	Stockout

**Table 5 ijerph-19-09232-t005:** Risk-sharing of partners under the option contract $\left(c_{0},c_{e},\;q_{O2}^{*},Q_{O2}^{*}\right)$.

Actual Demand	Manufacturer	Retailer
$x<q_{O2}^{*}=Q_{O2}^{*}$	Leftover	N/A
$q_{O2}^{*}=Q_{O2}^{*}<x$	N/A	Stockout

**Table 6 ijerph-19-09232-t006:** Risk-sharing of partners under the contract $\left(r,c_{0},c_{e},\;y_{A1}^{*},y_{O1}^{*},Q_{M1}^{*}\right)$.

Actual Demand	Manufacturer	Retailer
$x<y_{A1}^{*}$	Leftover	Leftover
$\;y_{A1}^{*}<x<y_{O1}^{*}\;+\;y_{A1}^{*}$	Leftover	N/A
$y_{O1}^{*}\;+\;y_{A1}^{*}<x<Q_{M1}^{*}$	Leftover	N/A
$y_{O1}^{*}\;+\;y_{A1}^{*}<Q_{M1}^{*}<x$	N/A	Stockout

**Table 7 ijerph-19-09232-t007:** Risk-sharing of partners under the PBO contract $\;(r,c_{0},c_{e},\;y_{A2}^{*},y_{O2}^{*},Q_{M2}^{*})$.

Actual Demand	Manufacturer	Retailer
$x<y_{A2}^{*}$	Leftover	Leftover
$\;y_{A2}^{*}<x<y_{O2}^{*}\;+\;y_{A2}^{*}=Q_{M2}^{*}$	Leftover	N/A
$y_{O2}^{*}\;+\;y_{A2}^{*}=Q_{M2}^{*}<x$	N/A	Stockout

## Data Availability

Not applicable.

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
