# Peer review of "Supply Chain Coordination under Carbon Emission Tax Regulation Considering Greening Technology Investment"

_ijerph, 2022, doi:10.3390/ijerph19159232_

Round 1
Reviewer 1 Report
Authors should check English grammar with a native speaker or a qualified philologist as English is not very accurate in the Manuscript. Also, it is recommended to Authors to summarize some postulates of their Manuscript as some of them do not mention both carbon emission tax and probability of investment into the greening technology. After these improvements paper will become shorter and easier to read too.
Reviewer 2 Report
At a time when ESG (environment, social contribution, and governance) is recently emphasized, the subject of this study, the carbon emission tax regulation, is very interesting and timely.
In particular, in recent years, most empirical studies have been mainly conducted. I think this study is different from other studies by providing proof of propositions.
However, in the case of such a theoretical study, there may be research limitations on whether it can be applied to actual companies or whether real cases can be presented.
In addition, since this study is based on 'one manufacture and one retailer case', different results may be presented in other situations.
Therefore, it will be necessary to describe that there are the research limitations on the matters pointed out above.
Reviewer 3 Report
Dear author/s,
many thanks for the opportunity to read your manuscript titled "Supply chain coordination under carbon emission tax regulation considering greening technology investment" submitting for International Journal of Environmental Research and Public Health.
The manuscript discusses about "supply chain coordination", particularly it explores the issues considering manufacturers invest in green technologies.
This study analysis these issues considering also some types of contracts to coordinate a manufacturer–retailer supply chain under carbon emission tax regulation.
The topic can be interesting, but the current version of the manuscript needs major revisions.
1-It is not clear the originality of the study. The authors should to better clarify it in the abstract and in the whole documento.
2-It is not clear the current gap in the literature to justify this study. The authors should to better clarify in the abstrcat, introduction, in the theoretical background.
3-Which is the major theory supporting this study? Pls, try to fill this major concern.
4-Where is the Discussion section? I would like to read a critical discussion about the results. You should include a section titled "discussion of results" in order to discuss the results under the theoretical lens. Which in the contribution provided your study to previous literature? Which is the ranking of this study compared to other studies?
5-In this document is not easy to identify the academics and practitional implications of the study. The authors should filled this lack.
I hope that the outcome of this review report does not discourage the authors from substantially reviewing this document.
Good Luck!
